# Ceramic resonators for targeted clinical magnetic resonance imaging of the breast

Alena Shchelokova [1], Viacheslav Ivanov[1], Anna Mikhailovskaya[1], Egor Kretov [1], Ivan Sushkov[2], Svetlana Serebryakova[3], Elizaveta Nenasheva[4], Irina Melchakova[1], Pavel Belov[1], Alexey Slobozhanyuk [1,6✉] & Anna Andreychenko [1,5,6]

Currently, human magnetic resonance (MR) examinations are becoming highly specialized with a pre-defined and often relatively small target in the body. Conventionally, clinical MR equipment is designed to be universal that compromises its efficiency for small targets. Here, we present a concept for targeted clinical magnetic resonance imaging (MRI), which can be directly integrated into the existing clinical MR systems, and demonstrate its feasibility for breast imaging. The concept comprises spatial redistribution and passive focusing of the radiofrequency magnetic flux with the aid of an artificial resonator to maximize the efficiency of a conventional MR system for the area of interest. The approach offers the prospect of a targeted MRI and brings novel opportunities for high quality specialized MR examinations within any existing MR system.

[1] Department of Physics and Engineering, ITMO University, Saint Petersburg 197101, Russia. [2] Department of Radiology, Vreden Russian Institute of Traumatology and Orthopedics, Saint Petersburg 195427, Russia. [3] All-Russian Center of Emergency and Radiation Medicine named after A.M. Nikiforov, Saint Petersburg 194044, Russia. [4] Ceramics Co., Ltd, Saint Petersburg 194223, Russia. [5] Research and Practical Clinical Center for Diagnostics and Telemedicine Technologies of the Moscow Health Care Department, Moscow 109029, Russia. [6]These authors contributed equally: Alexey Slobozhanyuk, Anna Andreychenko. ✉email: a.slobozhanyuk@metalab.ifmo.ru

As magnetic resonance imaging (MRI) provides outstanding quality and richness of the visualized information in combination with non-invasiveness and safety, MRI has gained significant medical use, and the amount of installed clinical magnetic resonance (MR) scanners in the world continues to grow[1]. Naturally, the amount of clinical applications of MRI is increasing, and with it, the number of targeted MR examinations of a particular body part or organ, e.g., the liver. The conventional concept of excitation and reception of the electromagnetic signal at clinical MR systems (i.e., 1.5–3 Tesla) includes a large, human-sized transmit radiofrequency (RF) volume body birdcage coil and receive-only surface coil (Fig. 1a). This combination provides acceptable image quality of MR examinations of the whole body as well as body parts or organs but has some limitations in the current state-of-the-art. These limitations are exceptionally prominent for a modern trend of highly specialized clinical MR investigations with a relatively small (compared to average body size) target (e.g., the head, spine, joint, breast)[2]. Such investigations, in principle, demand excitation and reception of MR signal only from the volume of interest. The RF magnetic field of the body birdcage coil is distributed over its total volume, requiring higher power during excitation[3] that limits the coil transmit efficiency for small areas. Furthermore, in the case of receive-only coils, common-mode currents are induced in RF cables by the body coil. Because of that reason, cable traps and matching circuits are included in receive coils designs to increase patient safety and signal-to-noise ratio[4]. In addition, the RF coils are usually relatively heavy and bulky. At the same time, the presence of fragile elements demands careful handling each time these coils are positioned and dismounted before and after every MR examination (Fig. 1a). The use of

dedicated transceive coils for extremities at the clinical MR systems has been proven useful to investigate fine details, e.g., of joint anatomies[5]. As the coils have to be placed directly onto the patient table of an MR system, their high RF power supply cables are thus close to a patient. The latter can potentially breach the safety of the procedure during the transmit stage. Also, the design of such coils restricts their only application to MR examinations of extremities.

Early disease diagnosis is an essential factor in increasing the chances for a successful outcome and in investigating preventative healthcare methods. In particular, early detection of breast cancer, which is the most common cancer type in women[6] and a death reason for every ninth woman, is crucial for survival. MRI provides the highest sensitivity to breast cancer, especially for younger women with dense glandular tissue but lacks specificity[7]. Recently, the specificity of MRI has been shown to increase employing advanced MR investigations[8,9], which rely on the high efficiency of the used MR equipment (mainly on its RF components) in the area of interest (e.g., the breast). Therefore, the performance of these advanced MR techniques is often unreliable or even unfeasible on a regular clinical MR scanner[10,11] that limits the application of these potentially valuable techniques for screening purposes.

Here we describe and demonstrate experimentally a universal concept of targeted clinical MRI (Fig. 1b) applied to breast imaging (Fig. 1c). The concept is based on a local redistribution and passive focusing of the RF magnetic field of the large body resonator (i.e., birdcage coil) through electromagnetic coupling with an additional subwavelength artificial dielectric resonator surrounding the target area. The dielectric resonator couples inductively to the body birdcage coil focusing the RF magnetic field within its cavity.

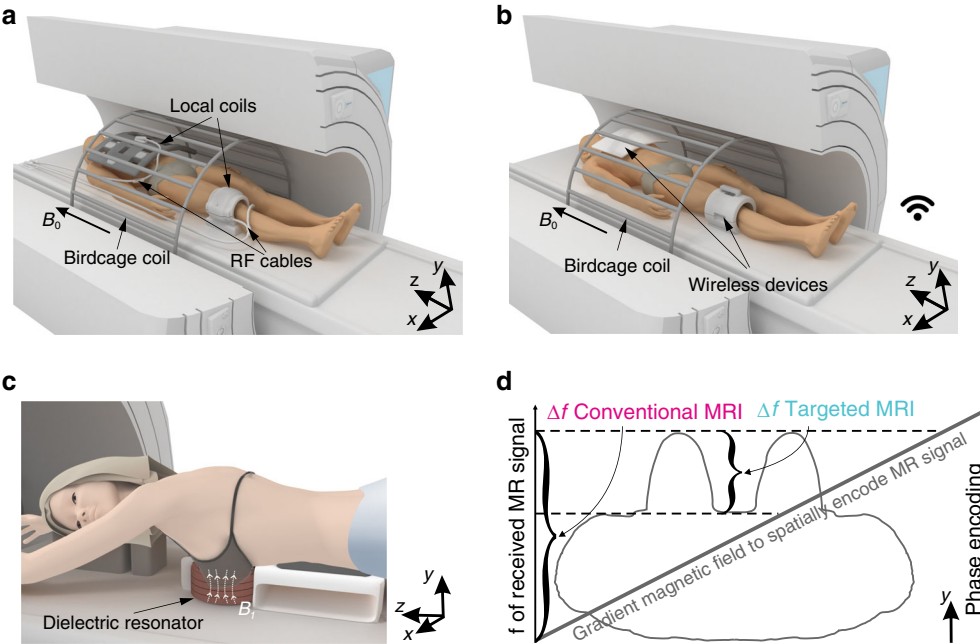

**Fig. 1 Conventional and targeted concepts of MRI. a** An illustration of a conventional acquisition scheme in the clinical MR systems. A body-sized RF resonator ("birdcage coil") placed in the wall of an MR system bore excites the MR signal, while multiple surface RF coils located directly on a patient ("local coils") receive MR signal and cabling system through a patient table delivers it to a spectrometer. $B_0$ indicates the direction of the static magnetic field of an MR system. **b** A targeted MRI concept. The same large body resonator excites and receives MR signal while a wireless device localizes both exciting and receiving RF magnetic fluxes of the body resonator to the region of interest. **c** An illustration of the actual experimental setup of targeted MRI for breast at 3 Tesla: the RF magnetic field ($B_1$-field) is confounded to the dielectric resonator cavity where a breast is located. **d** The gradient magnetic fields are always used during MR examinations to encode the MR signal through its frequency and phase. In the targeted MRI, the received signal bandwidth is limited to the size of the area of interest, while in the conventional MRI, the signal bandwidth is always proportional to patient body size. A significant reduction in the received signal bandwidth (in the phase-encoding direction) is beneficial for the total acquisition time of high-resolution imaging of the area of interest because of a decreased Nyquist frequency in the phase encoding direction.

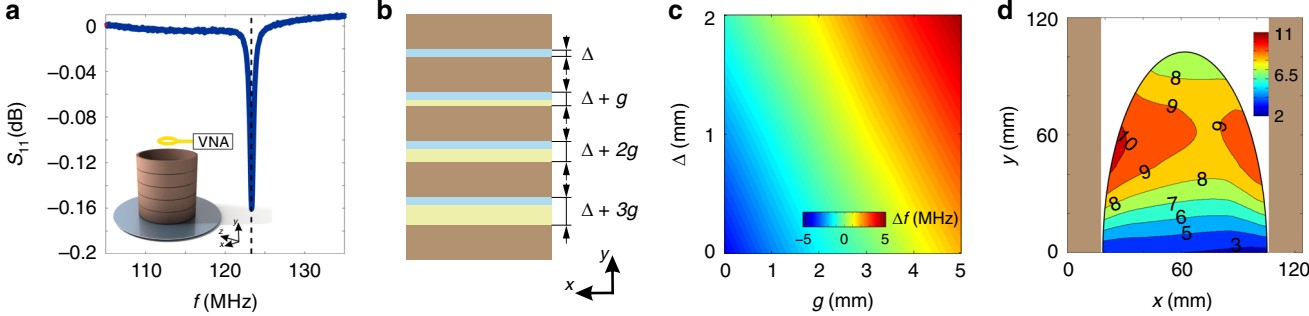

**Fig. 2 Resonator tuning principle and results of electromagnetic simulations. a** An illustration of the dielectric resonator prototype for targeted breast MRI at 3 Tesla. Correspondence of the device resonance frequency to the operational frequency of the used 3 Tesla clinical MR system (a vertical dashed black line) is shown with the $S_{11}$ measurements (blue curve). **b**, **c** The gaps between the discs of the resonator are used to tune the resonator to the operating frequency of a particular MR system. That makes the resonator universal. **d** An RF safety gain map inside the breast in the case of the body birdcage coil used in combination with the ceramic resonator relative to the birdcage coil alone. Different values and colors correspond to different RF safety gain values in different parts of the breast associated with inhomogeneities of root-mean-square value of the transmit RF electromagnetic field. Brown shaded areas show the boundaries of the resonator.

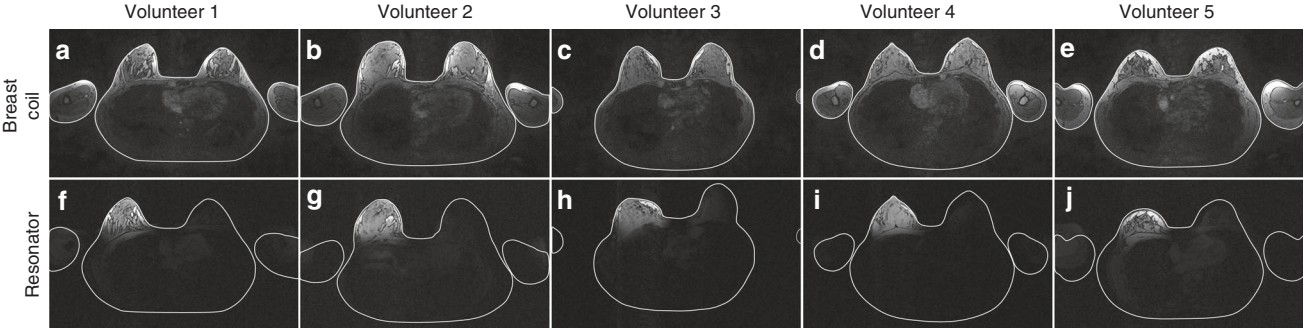

**Fig. 3 Experimental demonstration of targeted MR concept. a–e** MR images acquired for five healthy volunteers of BMI range 17.1–21 kg/m³ with the conventional methods: MR signal was excited through the body birdcage coil and received by 16-channel breast coil. **f–j** MR images acquired for the same volunteers with the transmitting/receiving birdcage coil together with the proposed prototype of the resonator.

In MRI, the focused excitation has a direct, advantageous impact on the image formation time and it is less prone to artifacts from the surrounding tissues and organs. The Nyquist criterion often restricts the latter in one (or two for three-dimensional (3D) acquisitions) of the spatial directions (a so-called phase-encoding direction). The targeted MRI limits the bandwidth of the acquired signal only to the region of interest (Fig. 1d), thus decreasing the Nyquist frequency and, as a result, the total acquisition time shrinks. There are alternative reconstruction methods that allow violation of the Nyquist criterion in order to speed up an MR acquisition[12–14]. On the other hand, they usually degrade MR image quality or their reconstruction time is unacceptably long for clinical workflow.

## Results

**Ceramic resonator design and electromagnetic simulation.** Dielectric materials are widely used in ultra-high field MRI with static field strength 7 Tesla (or higher) to enhance RF transmit efficiency locally and to reduce the specific absorption rate (SAR)[15–18]. However, with a few exceptions[19,20], at clinical field strengths, the influence of dielectric pads is limited due to the effect of non-resonant nature, whereas the dimensions of conventional dielectric resonators are too large to be practical. In this regard, novel artificial materials[21–27], e.g., dielectric metamaterials[28,29] and metasurfaces[30,31], can be employed to bring all-dielectric coils for practical MRI applications. Here, the targeted clinical MRI concept uses a artificial resonator, which consists of several dielectric discs with a very high relative

permittivity separated by specific gaps filled with a low dielectric constant material (Fig. 1c). The clinical MRI systems operate within a megahertz (MHz) frequency range, and 3 Tesla clinical systems work at about 120 MHz. Hence, to realize a sub-wavelength resonant structure at such a large wavelength, a material based on a mixture of $BaSrTiO_3$ doped with Mg was used[32]. This material is characterized by a relatively large permittivity value close to 1000 and a low level of dielectric loss measured at 1 MHz.

The resonator design is schematically shown in Fig. 2a (see details in "Methods"). We have employed the lowest frequency $TE_{01\delta}$ eigenmode and, by varying the height of the resonator and the spacing between the discs (Fig. 2b), we have tuned the eigenmode frequency to an operational frequency of the 3 Tesla MRI system (i.e., 123.25 MHz, see Fig. 2c). The mode magnetic field has a maximum value at the center of the resonator, while the electric field is zero[33] there. In addition, different separation of the dielectric discs allows fine-tuning the spectral response of the resonator without any additional lumped elements.

**Experimental demonstration of targeted MRI concept.** In vivo study was performed with five healthy volunteers of body index mass (BMI) range 17.1–21 kg m⁻³ for both conventional and targeted MRI concepts (Fig. 3). We have demonstrated that the resonator focuses the RF magnetic flux of a large body coil on the target, i.e., the breast (Fig. 3). The strong localization of the transmitting RF magnetic flux led to a 49 fold input power reduction on average compared with a conventional excitation

with the body birdcage coil alone (Table 1). As dissipated power proportional to the square of the electric field[34] during an MR examination with the proposed here targeted MRI concept, patient exposure to local RF electric fields is seven times lower. It was also confirmed by the numerical study (see details in "Methods") via RF safety calculations. RF safety was calculated using root-mean-square value of the transmitting magnetic field divided by a square root of the maximum local SAR for two cases: (1) in the presence of a resonator inside the birdcage coil and (2) with a birdcage coil used alone. The body birdcage coil RF safety gain increased by sevenfold on average across the breast in the presence of the dielectric resonator (Fig. 2d). At 3 Tesla MR systems, the average RF powers dissipated in patients are strictly regulated that often prolongs the total acquisition times of the sessions. Moreover, the presence of any metallic implants in the patient affects the parameters of the examinations, e.g., scans with the reduced RF power are prescribed even if the implant is placed outside the area under study[35]. The latest influences the quality of the resulting images. In the case of the targeted MRI, overall RF power can be reduced significantly, and it is deposited only in the region of interest. Therefore, the presence of metallic implants outside the area of examination would not restrict the MR examination with the proposed concept, achieving the optimum image quality without compromises on the RF safety of an examined subject (see Supplementary Fig. 2).

Moreover, by focusing a receiving RF magnetic flux of the body birdcage coil, the passive resonator proposed here made the birdcage coil an efficient receiver, comparable to a commercially available multi-channel breast coil (Fig. 3). However, due to the geometrical design of the resonator and the holder, some deviation in the breast shape image can be observed (see Fig. 3h,

j). This limitation can be avoided in future work with a curved shape put in the soft foam with special anatomical cuts.

In vivo breast imaging with the body birdcage coil coupled with the ceramic resonator demonstrates signal-to-noise ratio averaged over of five healthy volunteers comparable to a birdcage coil combined with state-of-the-art receive array (Fig. 4). It is worth mentioning that for relatively small breasts, where the loops of commercial coil relatively far from the investigated sample, i.e., breast coil works under suboptimal conditions, the resonator demonstrated even higher signal-to-noise ratio values (compare maps in Fig. 4d, e). Thus, in the concept of the targeted clinical MRI, a passive, wireless device placed next to the area of interest makes an MR examination safer (see Supplementary Fig. 2) and more time efficient, while using only the body birdcage coil to excite and receive signals. Next to it, the absence or a minimal excitation of the other body areas has additional advantages for targeted clinical MRI compared with the conventional MR examinations. Moving organs, such as the heart, e.g., due to the MRI acquisition principles, often create artifacts in the regions of interests and imaging strategies should be carefully selected to avoid such artifacts. In the targeted MRI, these artifacts are minimized, because hardly any signal is excited beyond regions of interest. Hence, the planning of an MR examination is easier and more robust to possible artifacts from the surrounding tissues (see Supplementary Fig. 3).

## Discussion
Here we have proposed the concept of targeted MRI using passive resonating devices wirelessly coupled to the main body coil of a clinical MR system. It offers promising opportunities to apply high-quality MR examinations within any existing MR system, which is especially advantageous for highly specialized MR examinations, such as breast MRI[36], pediatric MRI[37,38], and musculoskeletal MRI[39], demanding high RF powers and patient comfort and safety are particularly important.

A limitation of the current design is simplified cylindrical shapes of the ceramic elements. As a consequence, this prevents full coverage of fibroglandular tissue, in particular the breast coverage to the medial, dorsal, and lateral aspects, in some cases, maybe insufficient for a very comprehensive assessment. These limitations can be avoided in future work via fabrication of ergonomic design made of ceramic elements with a bigger inner diameter (see Supplementary Fig. 4) or even with a curved shape put in the soft foam with special anatomical cuts, which will be

**Table 1 The comparison of the signal-to-noise ratio (SNR) and the peak voltage ($U_{input}$)[a].**

| Setup | SNR (mean ± SD) | $U_{input}$, V (mean ± SD) |
|---|---|---|
| Breast coil | 69 ± 15 | 521 ± 23 |
| Resonator | 56 ± 11 | 74 ± 13 |

The comparison of the signal-to-noise ratio (SNR) and the peak voltage ($U_{input}$) of the RF pulse for the standard approach (birdcage coil and 16-channel breast coil) and the targeted concept (birdcage coil and dielectric resonator) averaged over five healthy volunteers.
[a]The square of the voltage amplitude is proportional to RF power.

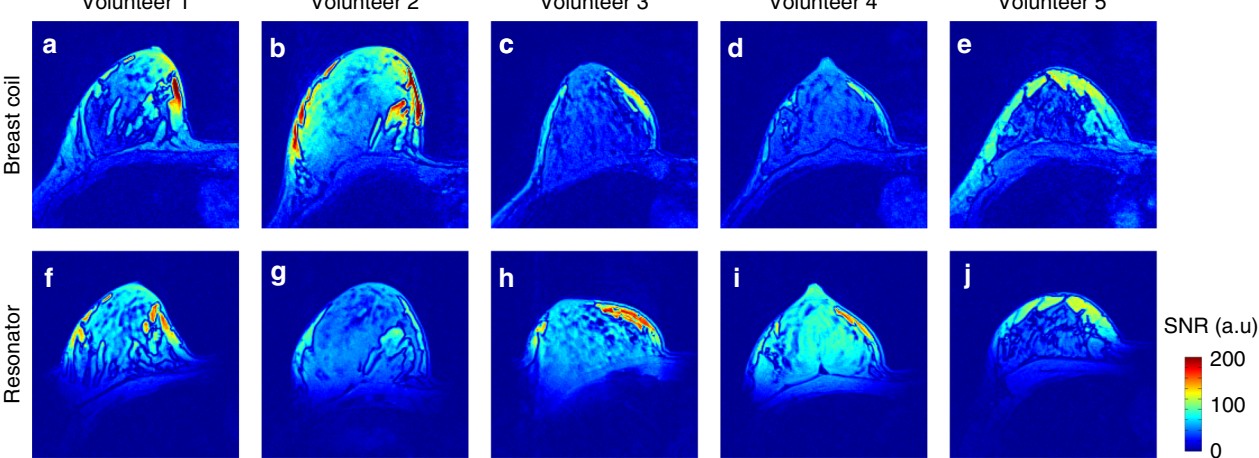

**Fig. 4 The comparison of the performances of the conventional and targeted MRI for a standard MR sequence via signal-to-noise ratio (SNR) maps.** During the conventional procedure (**a**–**e**), the MR signal was excited by a birdcage body coil and received by a 16-channel breast coil. In the targeted MRI concept (**f**–**j**), the birdcage coil was used for both transmitting and receiving in combination with the ceramic resonator.

better fitted with the anatomy of the body and the lymph nodes areas. It is also possible to combine the dielectric resonator with several loop coils. Such a combination could facilitate (in case needed) imaging not only fibroglandular tissue but also axillar areas without a need to move the patient. The technical approach of detuning dielectric resonators in the presence of loop coils was already demonstrated for 7 Tesla MRI[40]. Regarding the dedicated coils for sodium or other nucleus imaging, there are no limitations to use them with the dielectric resonator, as they operate at the different resonant frequencies and their interaction with the proposed resonator will be minimal.

The resonance conditions of the proposed structure strongly depend on the geometry of the high-permittivity elements. Any damage of the elements decreases its permittivity and may lead to the slight detuning of the structure from the resonant frequency, thus reducing resonator field focusing capability. It can be considered as a fail-safe mechanism, i.e., corrupted construction cannot be the reason for the dangerous field generation as it can happen with the standard clinical receive coils.

In conventional examinations, receive RF coils are detuned to prevent any risks associated with possible peak SAR values and resulted in burns of the tissues. That is why threefold safety mechanism[4,5] is realized in state-of-the-art receive coils. In the proposed concept of targeted MRI due to the transmit field focusing effect, the RF power amplitude of the transmit coil required to obtain optimal flip angle can be dramatically reduced. Therefore, although our resonator is active during the RF transmission (e.g., not detuned), the peak SAR value stays within limits and detuning methods (both active and passive) are not required in such conditions.

The ceramic resonator proposed here can be used with all existing MRI sequences, including fat suppression techniques (e.g., DIXON-based[9]). The only necessity could be an adjustment of the excitation RF pulse shapes for some sequences in order to correct the linear excitation RF field inhomogeneity across the breast. The latter has already been successfully realized in the high field clinical 7T scanners[41]. However, it is worth noting that for the most frequently used $T_1$-weighted gradient-echo (GRE) sequence in breast MRI (presented in this work), the RF field inhomogeneity was not an issue. Moreover, the proposed here targeted approach benefits the clinical application advanced techniques, e.g., diffusion-weighted imaging (DWI)[8] and chemical exchange saturation transfer (CEST)[9]. Both these methods rely on the high-amplitude RF pulses and are often performed in vivo under suboptimal conditions because of the RF safety concerns[10,11]. The proposed targeted approach allows for the safer in vivo application of these high-amplitude RF pulses because of the substantially reduced peak SAR values (see Supplementary Table 1), which creates optimal conditions for the clinical applications of the advanced MR sequences.

The targeted MRI can acquire high-resolution images without the associated linear (for two-dimensional MRI) and quadratic (for 3D MRI) time penalty, because a bandwidth of the received signal becomes proportional only to the region of interest excluding total body size (Fig. 1d). Relatively low time efficiency of an MR image formation is one of the limiting factors using MRI for population screening. The targeted MRI with its well-localized MR signal excitation and reception may become a needed step to introduce MRI as a noninvasive, safe, and highly informative screening tool for socially relevant diseases, especially when it is combined with the advanced MR acquisition schemes, such as MR fingerprinting[42]. Moreover, improvements in the resonator design, including excitation of non-radiative anapole modes[43] and involvement of novel mechanisms for achieving high Q-factors[44], will further improve the sensitivity of all-dielectric coils.

## Methods

**Resonator design.** The resonator was constructed from five ceramic discs, which are placed on a thin supporting plastic tube. Each disc was made from $BaSrTiO_3$, including Mg-containing compositions[32]. The relative dielectric permittivity of the disc material is $\varepsilon \sim 1000$ and $\tan \delta \sim 4 \times 10^{-4}$ (at 1 MHz). Each disc has the following dimensions: the inner diameter of 101.5 mm, the outer of 124.4 mm, and the height of 20 mm. The weight of the resonator (five ceramic discs and plastic spacers) is equal to 2.45 kg. An operational frequency of the resonator $TE_{01\delta}$ mode was tuned to 123.25 MHz (which corresponds to the proton Larmor frequency at the given 3 Tesla MR scanner) by changing the spacing between the ceramic discs. In general, spacers can be made from the low-permittivity and low-loss materials, such as plexiglas, plastic, or thick paper. Here we have used several thin discs made of plexiglas with relative dielectric permittivity around 3.5 and electrical conductivity of 0.02 S m$^{-1}$, and the height of 1.5 and 3 mm, and several discs made of paper with permittivity around 2.3 and the thickness of 0.25 mm. All spacers had the inner and outer diameters as for ceramic one and were placed at the same support tube between the ceramic discs. To preserve the efficiency of the dielectric resonator with proximity to the chest, a thin metallic disc made of the 14 μm-thick aluminum foil was added (shown in Fig. 2a). The metallic disc was sufficiently thin to avoid distortion of the gradient magnetic fields necessary for a spatial encoding of the MR signal. The aluminum foil was fixed at a supporting plastic tube via a plexiglas disc with an inner diameter of 101.5 mm and the outer diameter of 200 mm. It is noteworthy that metal had no contact with ceramics. The resonator prototype was placed inside an extruded polystyrene case of weight around 1 kg with two cylindrical holes to fix the resonator and provide the proper location of the volunteer and the breast inside the ceramic resonator during the MRI study.

The resonance frequency of the $TE_{01\delta}$ mode was measured using a small, non-resonant magnetic field probe placed above the dielectric resonator inside the MR system using a portable vector network analyzer (OBZOR TR1300/1, PLANAR LLC). Figure 2a illustrates that the minima of the probe $S_{11}$ curve corresponded to the desired frequency of 123.25 MHz.

To investigate the feasibility of $TE_{01\delta}$ mode frequency adjustment, we performed the electromagnetic numerical modeling in CST Microwave Studio 2017. The dielectric resonator was excited by a linearly polarized plane wave propagating along the $x$-direction with the magnetic field parallel to the $y$-axis (Fig. 2b). An operational frequency of the $TE_{01\delta}$ mode as a function of the gaps widths between the adjacent ceramic discs (parameters $\Delta$ and $g$ in Fig. 2b) was calculated using a parametric sweep. The metric $\Delta$ was varied from 0 to 2 mm, while g was changed between 0 and 5 mm. Thus, the total resonator height was varied within the range of 100–138 mm. Figure 2c demonstrates the feasibility to tune the operational frequency of the dielectric resonator ($\Delta f = f_{res} - 123.25$ MHz) in the range ±5 MHz. The ability to shift the operational frequency of the dielectric resonator slightly makes them applicable to any 3 Tesla MR systems.

**Numerical simulations.** Electromagnetic simulations were performed using a female voxelized model placed in the center of a standard body birdcage coil in CST Microwave Studio 2017 (see Supplementary Fig. 1). The body model was initially made for a supine position and the breasts did not have proper shape for the prone position. Therefore, an approximated layered breast model was created on top of the body model consisting of the following tissue types: skin ($\varepsilon = 72.93$, $\sigma = 0.49$ S m$^{-1}$), fat ($\varepsilon = 5.6$, $\sigma = 0.03$ S m$^{-1}$), gland ($\varepsilon = 67$, $\sigma = 0.8$ S m$^{-1}$). The dielectric resonator was placed around the right breast. For the reference case, the body birdcage coil was simulated with the same body model at the same central position, but without the resonator. The dielectric resonator coupled effectively to the birdcage coil and focused its $B_1^+$ field on the breast area (see Supplementary Fig. 2). The $B_1^+$ field and SAR averaged over 10 g tissue ($SAR_{av.10g}$) distributions were normalized to 1 W of total accepted power at the post-processing step.

A composite effect of the resonator on the birdcage coil transmit performance was evaluated by the ratio $\left|B_{1,rms}^+\right|/\sqrt{psSAR_{av.10g}}$ (RF safety, rms: root-mean-squared) in the presence of a resonator to the one with the body birdcage coil alone —a so-called RF safety gain:

$$\frac{\left|B_{1,rms}^+\right|/\sqrt{psSAR_{av.10g_{with resonator}}}}{\left|B_{1,rms}^+\right|/\sqrt{psSAR_{av.10g_{without resonator}}}},\qquad(1)$$

where the $\left|B_{1,rms}^+\right|$-field value was spatially averaged over the targeted area. The RF safety gain was calculated in the central axial slice (yx-plane) through the breast (Fig. 2d). The peak spatial SAR (10 g averaged) was increased by six times in the presence of the resonator (hot-spots depicted as black circles in Supplementary Fig. 2e, h), whereas an RF safety was increased by five- to ninefold across the breast (Fig. 2d). That means to obtain the same $B_1^+$-field value in the breast area and one should reduce the input power, thus reducing the peak values of SAR (see Supplementary Fig. 2f, i).

The International Electrotechnical Commission specifies peak SAR limits for normal and the first-level controlled operating modes of an MRI examination ($SAR_{av.10g} = 10$ W kg$^{-1}$ and $SAR_{av.10g} = 20$ W kg$^{-1}$, correspondingly). The peak SAR values are directly defined by the power accepted by the system ($P_{acc}$) that, in turn, sets the maximum of the RF magnetic field ($B_1^+$) amplitude. In Supplementary Table 1, a comparison of the $P_{acc}$ and corresponding mean $B_1^+$

values across the breast area for two operating modes without and with the resonator in place are presented. As could be seen in Supplementary Table 1, the mean $B_1^+$ values that could be reached with the resonator are more than sevenfold higher than with the birdcage coil alone. It means that (1) more efficient RF pulses could be used in DWI and CEST sequences; (2) repetition times of these sequences (that are often restricted by the RF safety regulations) could be shortened.

To improve the spatial coverage of the resonator, the design with a bigger inner diameter of ceramic discs (126 mm) was simulated. The results demonstrate a 1.6-fold higher field-of-view and a 1.13-fold higher RF magnetic field in the lateral areas (see Supplementary Fig. 4) in comparison with the proposed resonator with an inner diameter of 101.5 mm. However, it is worth noting that, in this case, a 30% loss in the RF magnetic field enhancement (mean value of the $B_1^+$-field) in the breast area was observed, i.e., the efficiency of the resonator operation is slightly decreased but still tenfold higher than compared with the body birdcage coil alone. Thus, it is worth noting that via engineering optimization of the design, one should compromise between the effectiveness of the resonator operation and field-of-view.

**MR experiments**. A written informed consent approved by the institutional review board was signed by volunteers before in vivo MR examinations. In vivo MR images of five healthy female volunteers of BMI range 17.1–21 kg m$^{-3}$ (min–max), with breast volume 381.4 ± 68.5 ml (mean ± SD) were acquired on 3T Siemens Magnetom Verio whole-body system using $T_1$-weighted 3D GRE sequence with Dixon-based fat suppression: FA = 15°, TR/TE$_1$/TE$_2$ = 7.8/2.4/3.7 ms, acquisition matrix = 640 × 380 × 208, and voxel size = 0.7 × 0.7 × 0.6 mm$^3$. A noise level was calculated as a SD of pixel values in the background areas of MR images. For comparison, MR images were acquired using the body birdcage coil and a 16-channel Breast Coil from Siemens. In the presence of the dielectric resonator, the birdcage coil was used for both transmission and reception of the MR signal. All MR experiments with resonator were performed only after precise RF power calibration procedure, which ensures that used input power within RF safety limits.

To confirm the effectiveness of the manual RF power calibration procedure, $B_1^+$ maps were evaluated using the double flip angle method ($\alpha = 20°$, $2\alpha = 40°$) with a small phantom filled with salted water, which imitated the breast, and a bigger one mimicking the human body (see Supplementary Fig. 3). These maps illustrate the actual flip angle distribution in the region of interest. $B_1^+$ stands for a circularly polarized RF magnetic flux field that is used to excite the MR signal and the flip angle depends directly onto $B_1^+$ amplitude. The GRE images to calculate a flip angle map were acquired with the following parameters: FA$_1$/FA$_2$ = 20/40, TR/TE = 6000/2.5 ms, acquisition matrix = 64 × 64, and a field of view = 320 × 320 mm$^2$. The resulting average angle was in both cases close to the nominal flip angle, confirming the reliability of the manual transmitter calibration.

**Reporting summary**. Further information on research design is available in the Nature Research Reporting Summary linked to this article.

## Data availability

The data that support the findings of this study are available from the corresponding author upon request. The authors entirely created Fig. 1 and Supplementary Fig. 1 based on the image of voxel model from commercially available software CST Microwave Studio.

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

## Acknowledgements

We are grateful to Professor Vladimir Fokin (Federal Almazov North-West Medical Research Center of Ministry of Healthcare of the Russian Federation) for advice regarding experiments with the dedicated breast coil, and to Professor Andrew Webb (Leiden University Medical Center) for useful discussions at preliminary stages of the project. The numerical calculations are supported by the Russian Foundation for Basic Research (grant number 18-32-20115). Experimental studies are supported by the Russian Science Foundation (Project Number 18-75-10088). A.A. acknowledges the support from the Government of the Russian Federation through the ITMO Fellowship and Professorship Program.

## Author contributions

A. Shchelokova, P.B., I.M., A. Slobozhanyuk, and A.A. conceptualized the work. A. Shchelokova, V.I., and A.M. performed electromagnetic simulations and data analysis. V.I., E.N., and E.K. carried out device fabrication and characterization. MRI experiments were designed and conducted by A. Shchelokova, A.A., V.I., I.S., and S.S. All authors discussed the results and commented on the manuscript.

## Competing interests

A patent has been filed with the title "Coil for magnetic resonance imaging of the breast" (# 2018146803 (27.12.2018)).
