## [Peer Review File · Nature Communications]

Peer Review File - Reviewers' comments first round:

Reviewer #1 (Remarks to the Author):

This work presents a novel approach to MRI with use of a strategically designed annular dielectric resonator coupled to the MRI system's body coil in order to greatly enhance signal in the region of the body inserted into the resonator, with application here to the human breast. Because conventional MRI typically relies on use of expensive, bulky, application-specific, and system-specific coils with heavy cables connected to the patient table, the potential to replace these coils with a passive resonator with no cables and which could work on MR systems from any supplier could greatly impact workflow, cost, and accessibility of MRI in the future. Presumably, the dielectric resonator would also be less susceptible to failure of electronics than conventional coils, adding to the potential benefits in terms of additional cost reduction and reliability. The authors demonstrate that their novel approach results in significantly lower power requirements (less heating of the patient) and much higher signal-to-noise ratio (higher image quality in the region of the body inserted into the resonator) than the conventional approach.

While the work is readily understandable, the grammar should be improved greatly before publication. The authors should seek input from someone skilled in technical writing in the English language, with special consideration of the relatively broad audience of Nature Communications. For example, added definition of "ultra-high field" MRI as referring to systems with static field strength greater than 7 Tesla, and either explicit relation of the voltages given in Figure 3 parts e and f to power deposition in the patient ($577^2/88^2=43$ times the power with the conventional application) or reporting power directly instead of voltage would be helpful.

It should be made more clear if the SNR distribution shown in Figure 3e is for excitation and reception with the body coil or excitation with the body coil and reception with the local coil. If it is for excitation and reception with the body coil, my enthusiasm based on improved SNR over the "conventional" approach would be greatly reduced.

While the current resonator is a prototype and demonstrates some clear potential advantages, some basic discussion of current limitations for clinical use would be appropriate. For example, it is my impression from the images shown that the current design may not readily allow for imaging of lymph nodes near the chest wall, which is often an important part of the breast exam.

Reviewer #2 (Remarks to the Author):

I would like to congratulate the authors to this well written and very interesting paper. In my opinion the novelty of the approach justifies a publication in nature communications very well. However, I see some details that can be improved (details below). I would thus suggest a revision before publication.

Claims

-The authors claim that by using a passive dielectric resonator the examination of human body extremities can be significantly improved over the current state of the art of human magnetic resonance imaging (MRI).

While the use of dielectric resonators is not novel in MRI the proposed use and design seems to be the first one producing not only very acceptable MR images but also a design that is useable in a clinical 3T MRI system.

-The authors further claim that a massive reduction in RF input power that is needed for the excitation of the spin system can be achieved using the novel approach. Also, the claim is made that massive improvements in RF safety and obtainable image information can be made with their approach.

While I can in principle agree with these claims (from a theoretical point of view) I do not see enough statistically sound evidence in the current version of the paper yet.

Major deficiency

I see a problem with the conventional coil that was used for comparison. Although no model is given, I think the authors have used a four channel siemens biopsy breast coil that compromises

of a flexible top coil. If that is the case, only two coil elements are used per breast configured as loops. The fact that this is a biopsy coil means that it is not a valid selection for a diagnostic setup as proposed by the novel setup here in this paper. Especially since a small breast is used basically you are receiving signal only from the top loop. The bottom loop will not contribute anything in case of small breasts. Also, the top loops sees naturally much more torso tissue. The reason for this is that in order to allow biopsy a loop configuration is used to be able to access the breast laterally for biopsy purposes. So overall, I think it is not the right coil for a fair comparison. I suggest to use a dedicated diagnostic breast coil (at least 4 receive elements for each breast). I personally would go for the Siemens breast 18 coil if you have that available. If another coil was used as I presume here, I apologise but non the less would like to hear a statement if the conventional coil is state of the art or not.

Minor Comments

I have some smaller comments that might aid the readability of the paper:

- On page one / two the authors say the number of clinical magnetic scanners is growing. That is right. Nonetheless a reference for this claim would be nice.
- On page two the sentence starting with "The RF magnetic field of the birdcage coil ..." might be rephrased. Also, it would be nice to have a citation for the claim that the standard MR imaging can only be conducted under suboptimal conditions. This claim should be either proven by literature-based evidence or dropped. I mean the authors are basically saying that the current state of the art of human MR imaging is suboptimal. I could interpret that the authors possibly meant that from a theoretical point this is not optimal. Anyhow rephrasing the sentence, might solve this problem easily.
- On page 5 "This material is characterized by a giant permittivity...". I would recommend to refrain from using superlative words like "giant" in a publication. I know a permittivity of 1000 is quite large compared to the permittivity of most materials but as it is well known that materials exist with an even larger permittivity. Just go for "high" would be my recommendation. Same for the conclusion on page 9.
- On page 6 please explain to the reader how a 43 fold reduction in power results in 8 times lower electric fields.
- Figure 3 page 8 right at the beginning the figure caption says "a,b,c Breast Images". I think some spaces are missing. I would recommend furthermore to rephrase so that is clear to the reader that only a and b are acquired with conventional methods. In fact if you read the whole sentence, caption and look at the figure it is clear but I think rephrasing would improve the quality.
- Figure 3 d) is not understandable to me. Neither the figure nor the accompanying text. I think either something is missing here or it is some other error in the description. Please add a scale with a unit to the figure (also for e and f) and rework the explanation please. What does the numbers mean?
- Figure 3 e and f. It seems to me that these are breasts from two different volunteers. However, the methods say only one volunteer was used. Please make sure that the same volunteer was used in the comparison. I would furthermore recommend to use immobilization of the breast or at least indicate if this was done or not. Also try to get at least a similar slice from the same breast using conventional and novel methodology. The reason for using the same volunteer is important as using different volunteer would change the load of the body coil. So, for the local coil I would assume a rather skinny volunteer based on the image while the dielectric resonator seems to have imaged a volunteer with a higher BMI. I do not want to ask for a full study here with all kinds of breast sizes and BMIs but think that a fair comparison is needed since strong claims are made. Please also include the number of volunteers. I strongly suggest to measure a statistically significant number of volunteers for the claims of reduced input powers etc and give mean values.

Apart from the points raised above I have some further questions:

- Can you comment on weight of the device and compare it with a conventional coil?
- If one of the resonator discs would get cracked. What would be the impact of that to the coil / images?
- how does the dielectric resonator perform on the axillar lymph nodes? While this is not part of the breast it is a place where often lesions are found and thus this area is important for breast

imaging. So, in other words, yes, it is nice to have reduced background signal from other tissue but not always. How much does the resonator see on the "outside"?

-How does the novel concept compare with a dedicated transmit/receive coil?

Regarding the reproducibility of the work presented here I am convinced that other researchers would be able to build a second resonator and confirm the results and claims made in this paper with the description given in the methods.

One more comment on RF safety:

The authors have investigated SAR in satisfactory manner. However, as I am sure the authors are also aware of, in conventional MRI one of the most feared risks are burns incurred by non-detuning receive loops. Thus, usually a threefold safety mechanism to prevent this is present per receive loop. It would be interesting to point that out and why this cannot happen with the novel concept presented here.

Once more, I would like to stress that besides the points raised above I really would like to see this work published. It gives a new perspective on how to tackle MRI and might lead to some rethinking in the field. In times when overall costs are rising in the global health care system I think novel and fresh approaches like this are needed in a multi-billion dollar industry.

From Reviewer #1: “Because conventional MRI typically relies on use of expensive, bulky, application-specific, and system-specific coils with heavy cables connected to the patient table, the potential to replace these coils with a passive resonator with no cables and which could work on MR systems from any supplier could greatly impact workflow, cost, and accessibility of MRI in the future. Presumably, the dielectric resonator would also be less susceptible to failure of electronics than conventional coils, adding to the potential benefits in terms of additional cost reduction and reliability”, and

From Reviewer #2: “I would like to congratulate the authors to this well written and very interesting paper. In my opinion the novelty of the approach justifies a publication in nature communications very well.

... I really would like to see this work published. It gives a new perspective on how to tackle MRI and might lead to some rethinking in the field. In times when overall costs are rising in the global health care system I think novel and fresh approaches like this are needed in a multi-billion dollar industry”.

We took all comments, critics, and suggestions very seriously, and have revised the paper substantially following the Reviewers reports.

We have addressed the additional comment of Reviewer #2 regarding the statistical evidence for the claimed advantages carefully. We performed a new set of experimental studies with five healthy volunteers of different body mass index and breasts content supporting the claimed advantages with statistical evidence. We have expanded the results summarized in Table 1 and revised Figures 3 and 4.

Moreover, we have compared the proposed ceramic resonator with a dedicated commercial breast coil (additional measurements were performed).

We would like to stress that the relevance of our research is the development of a cost-effective, fast 3-D breast MRI technology that potentially could speed up and enrich diagnostic information of a breast MR examination while ensuring the examination safety. In other words, the proposed device is focused on breast MR imaging as an effective, sensitive alternative to the established screening methods. Current limitations of our approach in imaging lymph nodes can be addressed in future work via fabrication of ceramic elements with a larger diameter or even with a curved shape, which will be better fitted with the anatomy of the body.

We do believe that the revision of the manuscript has brought this paper to the next level. This paper is of critical importance in the context of MRI because it presents a novel universal approach to perform targeted MRI examinations of the breast for screening purposes.

Please find below our point-to-point response to all concerns and comments raised by both Reviewers. Significant changes are highlighted in red in the revised main text.

We resubmit the paper for your further consideration.

Highest regards,
On behalf of all authors,
Dr. Alexey Slobozhanyuk

REPLY TO THE COMMENTS OF REVIEWERS

REVIEWER #1

While the work is readily understandable, the grammar should be improved greatly before publication. The authors should seek input from someone skilled in technical writing in the English language, with special consideration of the relatively broad audience of Nature Communications. For example, added definition of “ultra-high field” MRI as referring to systems with static field strength greater than 7 Tesla, and either explicit relation of the voltages given in Figure 3 parts e and f to power deposition in the patient ($577^2/88^2=43$ times the power with the conventional application) or reporting power directly instead of voltage would be helpful.

OUR REPLY

We thank the Reviewer for this comment. We revised the text carefully, fixing the grammar and including some definitions. We did not stress all the grammar changes but highlighted significant ones with red in the main text.

Total RF power is proportional to squared pulse amplitude times its duration [R. W. Brown et al., Magnetic Resonance Imaging: Physical Principles and Sequence Design, Second Edition, Wiley Blackwell, New Jersey 2014]. Since we are working with clinical systems, we are not aware of the form and duration of RF pulse, so we cannot estimate the total RF power directly. Nevertheless, we used the same pulses for both conventional and targeted concepts and compared the peak voltages that are available for the users on the clinical MR systems we used. Thus, a relative reduction of RF power deposition was estimated indirectly via squared relation of the peak voltages: on average 49 fold RF power reduction according to the new experimental results (new results summarized in Table 1).

REVIEWER #1

It should be made more clear if the SNR distribution shown in Figure 3e is for excitation and reception with the body coil or excitation with the body coil and reception with the local coil. If it is for excitation and reception with the body coil, my enthusiasm based on improved SNR over the “conventional” approach would be greatly reduced.

OUR REPLY

We have completely revised Figure 3, placing the explanation of conventional and targeted concepts in the caption. During the conventional procedure, the MR signal was excited by a birdcage coil and received by a 16-channel dedicated breast coil. In the targeted concept, the birdcage coil was used for both transmitting and receiving in combination with the dielectric resonator.

REVIEWER #1

While the current resonator is a prototype and demonstrates some clear potential advantages, some basic discussion of current limitations for clinical use would be appropriate. For example, it is my impression from the images shown that the current design may not readily allow for imaging of lymph nodes near the chest wall, which is often an important part of the breast exam.

OUR REPLY

Yes, we agree with the Reviewer that the current design does not fully facilitate an MR examination of some of the relevant lymph nodes. Nevertheless, the chest wall (and potential lymph nodes there) is visualized with the targeted concept.

Lymph nodes can be examined additionally after a quick screening procedure possible with proposed dielectric resonator if suspicious lesions in the breast itself are detected. The proposed

here technology, in contrast to the screening mammography, does not require painful breast compression, improving patient comfort and eliminating the risk of implant rupture in women with breast implants.

The limitations of current resonator prototype can be partially avoided in future work via fabrication of ceramic elements with a bigger diameter or even with a curved shape, which will be better fitted with the anatomy of the body including the lymph nodes areas.

We have added the descriptions of the prototype limitations with regard to the lymph nodes visualization in the Discussion part of the main text.

REVIEWER #2

The authors further claim that a massive reduction in RF input power that is needed for the excitation of the spin system can be achieved using the novel approach. Also, the claim is made that massive improvements in RF safety and obtainable image information can be made with their approach.

While I can in principle agree with these claims (from a theoretical point of view) I do not see enough statistically sound evidence in the current version of the paper yet.

OUR REPLY

To prove our claim of input RF power reduction, we performed statistical analyses with five volunteers of BMI range 17.1-21 kg/m³. The RF power level decreased by 49 times on average. These results are summarized in Table 1 of the main text.

REVIEWER #2

I suggest to use a dedicated diagnostic breast coil (at least 4 receive elements for each breast). I personally would go for the Siemens breast 18 coil if you have that available.

OUR REPLY

We have performed new experimental studies with five volunteers of BMI range 17.1-21 kg/m³ using 16-channel AI Breast Coil from Siemens for receiving MR signal. The coil elements are arranged in arrays of 6 elements for each breast, plus an axilla element and a cup design element on each side. The results are summarized in the revised version of Figure 3 (MR images) and Figure 4 (SNR-maps).

REVIEWER #2

Minor comments:

-On page one / two the authors say the number of clinical magnetic scanners is growing. That is right. Nonetheless a reference for this claim would be nice.

-On page two the sentence starting with “The RF magnetic field of the birdcage coil ...” might be rephrased. Also, it would be nice to have a citation for the claim that the standard MR imaging can only be conducted under suboptimal conditions. This claim should be either proven by literature-based evidence or dropped. I mean the authors are basically saying that the current state of the art of human MR imaging is suboptimal. I could interpret that the authors possibly meant that from a theoretical point this is not optimal. Anyhow rephrasing the sentence, might solve this problem easily.

- On page 5 “This material is characterized by a giant permittivity...”. I would recommend to refrain from using superlative words like “giant” in a publication. I know a permittivity of 1000 is quite large compared to the permittivity of most materials but as it is well known that materials exist with an even larger permittivity. Just go for “high” would be my recommendation. Same for the conclusion on page 9.

- On page 6 please explain to the reader how a 43 fold reduction in power results in 8 times lower electric fields.

- Figure 3 page 8 right at the beginning the figure caption says “a,b,c Breast Images”. I think some spaces are missing. I would recommend furthermore to rephrase so that is clear to the reader that only a and b are acquired with conventional methods. In fact if you read the whole sentence, caption and look at the figure it is clear but I think rephrasing would improve the quality.

OUR REPLY

We thank the Reviewer for these minor comments, helping us make the paper more clearly for readers. We have fixed all the issues in the main text.

REVIEWER #2

Figure 3 d) is not understandable to me. Neither the figure nor the accompanying text. I think either something is missing here or it is some other error in the description. Please add a scale with a unit to the figure (also for e and f) and rework the explanation please. What does the numbers mean?

OUR REPLY

Figure 3d (in the revised version 2d) is the result of electromagnetic simulations demonstrating an RF safety gain map in the central axial slice (yx-plane) through the breast in the case of the body-sized birdcage coil used in combination with the ceramic resonator relative to the birdcage coil alone. The evaluation of RF safety was described in Supplementary text. It was calculated using the root mean square value of the transmitting magnetic field divided by a square root of the maximum local specific absorption rate. Since the transmitting magnetic field (B_1^+) distribution across the resonator is not fully homogeneous and the breast consist of the tissues with different electrical conductivity values the RF safety gain of the body-sized resonator increased by 5 to 9 fold across the breast in the presence of the dielectric resonator.

We have improved the description of this image in the main text.

Regarding panels e and f (in the revised version Figure 4), they demonstrate measured SNR-maps with an arbitrary units scale.

REVIEWER #2

Figure 3 e and f. It seems to me that these are breasts from two different volunteers. However, the methods say only one volunteer was used. Please make sure that the same volunteer was used in the comparison. I would furthermore recommend to use immobilization of the breast or at least indicate if this, or not. Also try to get at least a similar slice from the same breast using conventional and novel methodology.

OUR REPLY

We revised Figure 3 according to new experimental data. MR images were acquired for five volunteers for conventional and targeted concepts (16 channel dedicated breast coil vs ceramic resonator). We tried to choose similar slices for each pair of images. However, since the

resonator case was rigid and did not have an ergonomic design unlike the breast coil case, the anatomical shape of the breast was a bit distorted in several studies. The development of an improved resonator case, which repeats the anatomical shape of the body, is also part of future work.

We have mentioned this limitation in the Discussion part of the main text.

REVIEWER #2

Can you comment on weight of the device and compare it with a conventional coil?

OUR REPLY

The weight of the resonator (five ceramic disks and plastic spacers) together with a holder (case) is equal to 3.45 kg. In the case of using two resonators for bilateral imaging, the weight will increase up to 6 kg. This value is comparable and even less than the weight of the standard breast coil of 7.4 kg.

We have included the information about the resonator weight in Supplementary text (Methods).

REVIEWER #2

If one of the resonator discs would get cracked. What would be the impact of that to the coil / images?

OUR REPLY

Resonance conditions of the proposed structure strongly depend on the geometry of the high-permittivity elements. Any damage of the elements may decrease its permittivity and it may lead to the slight detuning of the structure from the resonant frequency, thus reduce resonator field focusing capability. It can be considered as a fail-safe mechanism i.e. corrupted construction cannot be the reason for the dangerous field generation as it can happen with the standard clinical receiving coils. However, we have already performed a large number of experiments and small damages (e.g. small chips) did not affect the resonator efficiency.

REVIEWER #2

How does the dielectric resonator perform on the axillar lymph nodes? While this is not part of the breast it is a place where often lesions are found and thus this area is important for breast imaging. So, in other words, yes, it is nice to have reduced background signal from other tissue but not always. How much does the resonator see on the “outside”?

OUR REPLY

The current design does not facilitate an MR examination of some of the relevant lymph nodes. Nevertheless, the chest wall (and potential lymph nodes there) is visualized with the targeted concept.

Lymph nodes can be examined additionally after a quick screening procedure possible with proposed dielectric resonator if suspicious lesions in the breast itself are detected. The proposed here technology, in contrast to the screening mammography, does not require painful breast compression, improving patient comfort and eliminating the risk of implant rupture in women with breast implants.

The limitations of current resonator prototype can be partially avoided in future work via fabrication of ceramic elements with a bigger diameter or even with a curved shape, which will be better fitted with the anatomy of the body including the lymph nodes areas.

We have added the descriptions of the prototype limitations with regard to the lymph nodes visualization in the Discussion part of the main text.

REVIEWER #2

How does the novel concept compare with a dedicated transmit/receive coil?

OUR REPLY

In theory, a transmit/receive coil is the most appropriate solution for imaging of the small regions of the human body. However, there is no such coil for 3T breast imaging. The quality of MR images obtained with the dielectric resonator coupled with body-sized birdcage coil are comparable with ones obtained by 16-channels receive-only coil from Siemens (it was confirmed by our experiments in the revised version of the paper).

Moreover, the development of the clinical MRI equipment aims for universality, which can be provided by employing the body-sized birdcage coil as the source and dedicated local coils to receive an MR signal. As a consequence, the amount of the available Tx/Rx channels in clinical systems is limited and dedicated transmit/receive coils are used rarely. Known models of the transmit/receive breast coils are specific and compatible with specialized MRI scanners only, (<https://www.auroramri.com/>). In addition, the absence of high-power supply cables and improvement of RF safety per exposed body part (<https://doi.org/10.1002/mrm.24690>) are additional benefits of the proposed concept in comparison with the local transmit/receive coils.

REVIEWER #2

Regarding the reproducibility of the work presented here I am convinced that other researchers would be able to build a second resonator and confirm the results and claims made in this paper with the description given in the methods.

OUR REPLY

We have expanded the section about the resonator design in Methods.

REVIEWER #2

The authors have investigated SAR in satisfactory manner. However, as I am sure the authors are also aware of, in conventional MRI one of the most feared risks are burns incurred by non-detuning receive loops. Thus, usually a threefold safety mechanism to prevent this is present per receive loop. It would be interesting to point that out and why this cannot happen with the novel concept presented here.

OUR REPLY

We are grateful to this comment of the Reviewer. In conventional examinations receive coils are detuned to prevent any risks associated with possible peak SAR values and resulted burns of the tissues. That is why threefold safety mechanism is realized in state-of-the-art receive coils. In the proposed concept of targeted MRI due to the transmit field focusing effect the RF power amplitude of the transmit coil required to obtain optimal flip angle can be dramatically reduced. Therefore, while our resonator is active during the RF transmission (e.g. not detuned) the peak SAR value stays within the limits and detuning methods (both active and passive) are not required in such conditions.

Additionally, since the resonance conditions of the proposed structure strongly depend on the geometry of the high-permittivity elements. Any damage of the elements decreases its permittivity and may lead to the slight detuning of the structure from the resonant frequency, thus reducing resonator field focusing capability. It can be also considered as a fail-safe mechanism i.e. corrupted construction cannot be the reason for the dangerous field generation as it may happen with the standard clinical receiving coils.

Reviewers' comments second round:

Reviewer #1 (Remarks to the Author):

All my concerns have been adequately addressed. The manuscript is significantly improved.

Reviewer #2 (Remarks to the Author):

I would like to congratulate the authors to the revised version of the paper. My remarks from the first iteration of the paper have been fully answered in a satisfactory manner. I think the paper is on good track to being published. I overall recommend publication.

Minor comments:

Page 2, line 34: Rephrase sentence.

My suggestion would be "As magnetic resonance imaging (MRI) provides outstanding quality and richness of the visualized information in combination with non-invasiveness and safety, MRI has gained significant medical use and the amount of installed clinical magnetic resonance (MR) scanners in the world continues to grow."

Page 2, line 37: Rephrase sentence.

My suggestion would be "Naturally, the amount of clinical applications of MRI is increasing and with it the number of targeted MR examinations of a particular body part or organ like for example the liver."

Page 2, line 39: Rephrase sentence.

My suggestion would be "The conventional concept... the electromagnetic MR-signal..."

Page 2, line 42: Rephrase sentence.

My suggestion would be "This combination provides acceptable image quality or MR examinations of the whole body as well as body parts or organs but has some limitations in the current design in the state of the art."

Page 2, line 46/467: add word.

My suggestion would be "The RF magnetic field of the birdcage body coil..."

Page 2, line 50: Rephrase sentence.

My suggestion would be "The use of dedicated excitation coils for extremities at the clinical MR systems that have been proven useful to investigate fine details for example of joint anatomies"

Page 2, line 51: ambiguously correct.

Only in transmit coils the cables of the coils are high power RF power supply cables. It is true that this can be a safety issue however, in most of the cases and also in the case of the scenario described in the context of this publication it seems to me more appropriate to talk about receive coil RF cables. Those also possess safety hazards but due to induced standing wave effects by the electromagnetic excitation field of the body coil and not by the power that is flowing within the receive cables. Because of that reason cable traps are included. I suggest to add this information and add a reference to the topic of cable traps.

Page 3, line 63: semantics.

The resonator is said to "localize" the RF field of the body coil within the dielectric resonator. I wonder if "focusing" might not be a better fitting word.

Page 3, line 66: semantics.

swap "recovery" with "outcome".

Page 3, line 68: missing reference.

"...which is the most common cancer type in women..." true, but please add a reference to that sentence. I suggest WHO cancer statistics for example.

Page 3, line 73: missing reference.

Please provide a reference for the claim that "...the performance of these advanced MR techniques is often unreliable or even unfeasible on a regular clinical MR scanner...".

Reviewer #3 (Remarks to the Author):

I thank the authors for their careful revision of the manuscript and for addressing the questions raised by the reviewers. As I am a medical doctor (radiologist) and not a physicist or engineer I refrain from commenting on these aspects of your work and leave the assessment to the specialists in the field, however since being asked to specifically address the clinical point of view on the suggested innovative approach of using novel resonators to improve breast MR imaging I would suggest to include additive information:

1)As previously mentioned by the reviewers for breast cancer diagnostics it is of high relevance to a) cover the entire breast to the chest wall and b) especially the lateral parts of the FGT (in an optimal case up to the axillary region in order to assure not missing suspicious lesion there). From a practical point of view with your technique it would thus currently be necessary to re-examine the women with a "standard" coil to obtain the rest of images covering all areas of interest which would probably interfere with the aim of saving time (re-placing the patient for the examination is quite time consuming and re-placing a patient causes issues with regards to image co-localization especially for breast imaging), so how realistic is it to cover these lateral, dorsal and medial aspects in future work?

a. Herein you mention to the reviewer that your current design covers the chest wall, and while I agree that your images demonstrate, that some peaks of the image reach the chest wall it is very obvious that the breast coverage to the medial, dorsal and lateral aspects is insufficient for a comprehensive assessment (you can see that substantial amounts of FGT are not depicted as compared to the breast coil), this limitation might benefit from being slightly more emphasized in the discussion for the clinicians reading the manuscript.

2)Your setup (Suppl. Fig 1) seems to be already pushed to the edge of the chest wall quite strictly, so how could it be possible to increase imaged volume towards the chest wall further from a practical point of view (to what degree do you need to maintain the perfect cylindrical shape of the resonator for imaging?)?

3)Are there any limitations with regards to the applied imaging sequences or is the approach usable in the same manner for all existing MR imaging sequences (all routine clinical sequences and upcoming sequences e.g. DWI, CEST, what about techniques that currently need dedicated coils such as sodium imaging)?

4)To what degree does the technique influence homogeneity and artefacts commonly observed at the inner borders of the breast close to the sternum (does it avoid or increase such artifacts)?

5)Since women will be in need to be placed in some sort of "coil shaped structure" in prone position anyhow for the breast MRI examination to what degree could your approach be combined with existing coil design enhancing the potential even more?

6)Safety of the MRI examination is of utmost importance, so are there any potential emerging safety issues with such a technique that we as medical doctors should anticipate, e.g. with tattoos or breast implants? From your supplemental Figure 2 it seems your SAR peak is almost 10 fold higher (scales are different in between the images, this should be corrected) with a spatial focus on the skin, to what degree does that influence to applicability of the method?

7)Do fat suppression techniques interfere with this approach or increase the issue with locally insufficient fat suppression problems in breast imaging?

8)Novel sequences (e.g. TWIST VIBE) can acquire images of the full breast in a couple of seconds only, so would it be possible to use your approach to increase resolution instead of reducing imaging time as well?

9) A "breast coil-shaped" device will be needed for the placement of women in prone position anyhow as well with your device, so what is the "pure" remaining clinical advantage [besides the important factor of costs for the breast coil] – to be asking this as a "provocative" question that might raise at some point when you aim at a clinical introduction of your technique?

REPLY TO THE COMMENTS

REVIEWER #2

Minor comments.

OUR REPLY

We have revised our manuscript following by all minor comments from Reviewer #2.

REVIEWER #3

I thank the authors for their careful revision of the manuscript and for addressing the questions raised by the reviewers. As I am a medical doctor (radiologist) and not a physicist or engineer I refrain from commenting on these aspects of your work and leave the assessment to the specialists in the field, however since being asked to specifically address the clinical point of view on the suggested innovative approach of using novel resonators to improve breast MR imaging I would suggest to include additive information:

OUR REPLY

We are grateful to the Reviewer for a positive evaluation of our work and essential questions and comments, which allows us to improve our manuscript.

REVIEWER #3

1)As previously mentioned by the reviewers for breast cancer diagnostics it is of high relevance to a) cover the entire breast to the chest wall and b) especially the lateral parts of the FGT (in an optimal case up to the axillary region in order to assure not missing suspicious lesion there). From a practical point of view with your technique it would thus currently be necessary to re-examine the women with a “standard” coil to obtain the rest of images covering all areas of interest which would probably interfere with the aim of saving time (re-placing the patient for the examination is quite time consuming and re-placing a patient causes issues with regards to image co-localization especially for breast imaging), so how realistic is it to cover these lateral, dorsal and medial aspects in future work?

a. Herein you mention to the reviewer that your current design covers the chest wall, and while I agree that your images demonstrate, that some peaks of the image reach the chest wall it is very obvious that the breast coverage to the medial, dorsal and lateral aspects is insufficient for a comprehensive assessment (you can see that substantial amounts of FGT are not depicted as compared to the breast coil), this limitation might benefit from being slightly more emphasized in the discussion for the clinicians reading the manuscript.

OUR REPLY

This is a very important point. All concerns mentioned above can be definitely solved in future work, adapting the current resonator design for real practice, while using the novel targeted approach proposed in our work. In particular, to obtain full coverage of FGT, the following modifications can be done:

1) the usage of ceramic rings with a larger inner diameter, e.g., 20% larger in comparison with the current design, to improve spatial coverage of the resonator. In order to confirm this idea, we performed an additional numerical simulation with this design, which demonstrates a 1.6-fold larger field-of-view and a 1.13-fold higher RF magnetic field in the lateral areas (see Fig. S1a,b below). However, it is worth noting that in this case, a 30% loss in the RF magnetic field enhancement in the breast area is possible. However, compared to the body coil alone the resonator with the larger diameter still enhances the RF magnetic field by 10-folds. So, via

engineering optimization of the design, one should compromise between the effectiveness of the resonator operation and field-of-view.

2) it is also possible to combine the dielectric resonator with several loop coils. Such a combination could facilitate (in case needed) imaging not only FGT but also axillary areas without a need to move the patient. The technical approach of detuning dielectric resonators in the presence of the loop coils was recently demonstrated for 7T MRI [https://www.sciencedirect.com/science/article/abs/pii/S1090780717302392]. We have included related discussions and new numerical results in the revised version of the manuscript.

Added to the main text:

“A limitation of the current design is simplified cylindrical shapes of the ceramic elements. As a consequence, this prevents full coverage of fibroglandular tissue, in particular, the breast coverage to the medial, dorsal and lateral aspects in some cases may be insufficient for a very comprehensive assessment. These limitations can be avoided in future work via fabrication of ergonomic design made of ceramic elements with a bigger inner diameter (see Supplementary Fig. 4), or even with a curved shape put in the soft foam with special anatomical cuts, which will be better fitted with the anatomy of the body and the lymph nodes areas. It is also possible to combine the dielectric resonator with several loop coils. Such a combination could facilitate (in case needed) imaging not only fibroglandular tissue but also axillar areas without a need to move the patient. The technical approach of detuning dielectric resonators in the presence of loop coils was already demonstrated for 7 Tesla MRI⁴⁰.”

“To improve the spatial coverage of the resonator, the design with a bigger inner diameter of ceramic disks (126 mm) was simulated. The results demonstrate a 1.6-fold higher field-of-view and a 1.13-fold higher RF magnetic field in the lateral areas (see Supplementary Fig. 4) in comparison with the proposed resonator with an inner diameter of 101.5 mm. However, it is worth noting that, in this case, a 30% loss in the RF magnetic field enhancement (mean value of the B_1^+ -field) in the breast area was observed, i.e., the efficiency of the resonator operation is slightly decreased but still 10-fold higher than compared to the birdcage body coil alone. Thus, it is worth noting that via engineering optimization of the design, one should compromise between the effectiveness of the resonator operation and field-of-view”.

Added to Supplementary Information:

Fig. S4. Numerical simulation results of the transmitting magnetic field (B_1^+) for (a) the proposed ceramic resonator and (b) the resonator with 20% higher inner diameter. Brown shaded areas show the boundaries of the resonator. White dashed lines indicate the field-of-view improvement, where the mean value of the B_1^+ -field was calculated.

REVIEWER #3

2) Your setup (Suppl. Fig 1) seems to be already pushed to the edge of the chest wall quite strictly, so how could it be possible to increase imaged volume towards the chest wall further from a practical point of view (to what degree do you need to maintain the perfect cylindrical shape of the resonator for imaging)?

OUR REPLY

As was mentioned above (please, see the answer to the comment #1), to increase the imaged volume, it is possible to increase the inner diameter of the resonator. Theoretically, there are other options for the resonator design, including spatially curved shapes, which would better fit the patient's anatomy. The general restriction to the design process is that the RF magnetic field created by displacement currents in ceramic parts and better performance can be achieved only when they surround the region of interest. From that point of view, cylindrical shape is an ideal symmetric structure for investigation. Thus, a cylindrical design is a starting point of the non-trivial optimization problem, which will be solved in our future work.

Added to the main text:

“These limitations can be avoided in future work via fabrication of ergonomic design made of ceramic elements with a bigger inner diameter (see Supplementary Fig. 4), or even with a curved shape put in the soft foam with special anatomical cuts, which will be better fitted with the anatomy of the body and the lymph nodes areas.”

REVIEWER #3

3) Are there any limitations with regards to the applied imaging sequences, or is the approach usable in the same manner for all existing MR imaging sequences (all routine clinical sequences and upcoming sequences, e.g., DWI, CEST, what about techniques that currently need dedicated coils such as sodium imaging)?

OUR REPLY

The ceramic resonator proposed here can be used with all existing MRI sequences. The only necessity could be an adjustment of the excitation radiofrequency (RF) pulse shapes for some sequences in order to correct the linear excitation RF field inhomogeneity across the breast. The latter has already been successfully realized in the high field clinical 7T scanners [<https://onlinelibrary.wiley.com/doi/full/10.1002/mrm.23264>]. However, it is worth noting that for the most frequently used T1w GRE sequence in breast MRI (presented in this work), the RF field inhomogeneity was not an issue.

Concerning DWI and CEST, the proposed targeted approach benefits the clinical application of these advanced techniques. Both these methods rely on the high amplitude RF pulses and are often performed *in vivo* under suboptimal conditions because of the RF safety concerns. The proposed here targeted approach allows for the safer *in vivo* application of these high amplitude RF pulses because of the substantially reduced peak SAR values, which creates optimal conditions for the clinical applications of the advanced MR sequences. From the RF safety gain, it follows that while remaining under the same RF safety conditions, RF pulses amplitudes could be increased at least by 5-fold compared to the conventional birdcage body coil.

The International Electrotechnical Commission specifies peak SAR limits for normal and the first-level controlled operating modes of an MRI examination ($SAR_{av,10g}=10$ W/kg and $SAR_{av,10g}=20$ W/kg, correspondingly). The peak SAR values are directly defined by the power accepted by the system (P_{acc}) that, in turn, sets the maximum of the RF magnetic field (B_1^+) amplitude. In Table S1, we present a comparison of the P_{acc} and corresponding mean B_1^+ values across the breast area for two operating modes without and with the resonator in place. As could

be seen in Table S1, the mean B_1^+ values that could be reached with the resonator are more than 7-fold higher than with the birdcage coil alone. It means that (1) more efficient RF pulses could be used in DWI and CEST sequences; (2) repetition times of these sequences (that are often restricted by the RF safety regulations) could be shortened.

Regarding the dedicated coils for sodium or other nucleus imaging, there are no limitations to use them with the proposed resonator, since they operate at the different resonant frequencies, and their interaction with the resonator will be minimal.

Added to the main text:

“The ceramic resonator proposed here can be used with all existing MRI sequences. The only necessity could be an adjustment of the excitation RF pulse shapes for some sequences in order to correct the linear excitation RF field inhomogeneity across the breast. The latter has already been successfully realized in the high field clinical 7T scanners⁴¹. However, it is worth noting that for the most frequently used T_1 -weighted gradient-echo (GRE) sequence in breast MRI (presented in this work), the RF field inhomogeneity was not an issue. Moreover, the proposed here targeted approach benefits the clinical application advanced techniques, e.g. diffusion weighted imaging (DWI)⁸ and chemical exchange saturation transfer (CEST)⁹. Both these methods rely on the high amplitude RF pulses and are often performed *in vivo* under suboptimal conditions because of the RF safety concerns^{10,11}. The proposed targeted approach allows for the safer *in vivo* application of these high amplitude RF pulses because of the substantially reduced peak SAR values, which creates optimal conditions for the clinical applications of the advanced MR sequences”.

“Regarding the dedicated coils for sodium or other nucleus imaging, there are no limitations to use them with the dielectric resonator, since they operate at the different resonant frequencies, and their interaction with the proposed resonator will be minimal”.

Added to methods:

“The International Electrotechnical Commission specifies peak SAR limits for normal and the first-level controlled operating modes of an MRI examination ($SAR_{av,10g}=10$ W/kg and $SAR_{av,10g}=20$ W/kg, correspondingly). The peak SAR values are directly defined by the power accepted by the system (P_{acc}) that, in turn, sets the maximum of the RF magnetic field (B_1^+) amplitude. In Table S1 (Supplementary Information), a comparison of the P_{acc} and corresponding mean B_1^+ values across the breast area for two operating modes without and with the resonator in place are presented. As could be seen in Table S1 (Supplementary Information), the mean B_1^+ values that could be reached with the resonator are more than 7-fold higher than with the birdcage coil alone. It means that (1) more efficient RF pulses could be used in DWI and CEST sequences; (2) repetition times of these sequences (that are often restricted by the RF safety regulations) could be shortened.”

Added to Supplementary Information:

Table S1 | The comparison of the P_{acc} and corresponding mean B_1^+ values across the breast area for normal and the first-level controlled operating modes of an MRI examination without and with the resonator in place.

Operating mode		Without resonator	With resonator
Normal ($SAR_{av,10g}=10$ w/kg)	P_{acc}^*	66 W	17 W
	mean B_1^+ in the breast	1 uT	8 uT
First-level ($SAR_{av,10g}=20$ w/kg)	P_{acc}^*	132 W	34 W
	mean B_1^+ in the breast	1.5 uT	11.7 uT

* P_{acc} – power accepted by the system.

REVIEWER #3

4) To what degree does the technique influence homogeneity and artefacts commonly observed at the inner borders of the breast close to the sternum (does it avoid or increase such artifacts)?

OUR REPLY

These artifacts are caused by the main field (B_0) inhomogeneities. The resonator proposed here has no prominent influence on the B_0 field distribution. Thus, the mentioned artifacts will remain the same in the presence of the resonator as with the conventional set-up.

REVIEWER #3

5) Since women will be in need to be placed in some sort of “coil shaped structure” in prone position anyhow for the breast MRI examination to what degree could your approach be combined with existing coil design enhancing the potential even more?

OUR REPLY

As was mentioned above (please, see the answer for the comment #1), making the resonator detunable and/or decoupled (e.g., geometrically) from the local receive coils will only increase the advantage of the proposed here technique because of the increased number of receive channels.

Added to the main text:

“It is also possible to combine the dielectric resonator with several loop coils. Such a combination could facilitate (in case needed) imaging not only fibroglandular tissue but also axillar areas without a need to move the patient. The technical approach of detuning dielectric resonators in the presence of loop coils was already demonstrated for 7 Tesla MRI⁴⁰”.

REVIEWER #3

6) Safety of the MRI examination is of utmost importance, so are there any potential emerging safety issues with such a technique that we as medical doctors should anticipate, e.g. with tattoos or breast implants? From your supplemental Figure 2 it seems your SAR peak is almost 10 fold higher (scales are different in between the images, this should be corrected) with a spatial focus on the skin, to what degree does that influence to applicability of the method?

OUR REPLY

Indeed, for the same input power, the peak SAR value in the case of the ceramic resonator is higher. However, for the quality of an MRI examination, the level of RF magnetic field (B_1^+) is of primary importance, while the peak SAR values influence only the RF safety of the procedure and have no consequences for its clinical quality. Thus, it is essential not to consider the peak SAR values alone but in combination with the B_1^+ values, i.e., RF safety ($B_1^+/\sqrt{\text{SAR}}$). The RF safety of the resonator is a 7-fold higher on average than in comparison with the birdcage body coil (see Fig. 2d in the main text). We performed additional SAR calculations keeping the mean RF magnetic field value in the breast area the same (e.g., mean $B_1^+=0.14$ uT) with and without the resonator. As can be seen in the supplementary Figure 2f,i, the peak SAR value in the presence of the resonator was 40 times lower.

Concerning the tattoos and breast implants, no additional safety issues are expected besides the conventional ones. However, an essential benefit of the proposed resonator is the control of the spatial location of the peak SAR values. The highest SAR values are located within the region of interest with the proposed method, while with the birdcage body coil; their location is much less predictable.

Added to the main text:

“That means to obtain the same B_1^+ -field value in the breast area, and one should reduce the input power, thus reducing the peak values of SAR (see Supplementary Fig. 2f,i)”.

Added to Supplementary Information:

Fig. S2. Numerical simulation results of the $|B_{1,rms}^+|$ -field, and $SAR_{av,10g}$ distributions for a human voxel model placed inside the birdcage body coil. The calculated $|B_{1,rms}^+|$ maps without (a), and with (b) the resonator for 1 W of total excitation power, and (c) with the resonator for 0.005 W, to create the same mean value of $|B_{1,rms}^+|$ in the breast as for the reference case (the birdcage body coil alone). The calculated $SAR_{av,10g}$ maps: (d,g)—reference case without the resonator for 1 W of accepted power ($|B_{1,rms}^+|=0.14$ uT in the breast area); (e),(h)—with the resonator for the same accepted power 1 W as in the reference case ($|B_{1,rms}^+|=2$ uT); (f),(i)—with the resonator for the same $|B_{1,rms}^+|$ as in the reference case (accepted power 0.005 W). White solid lines indicate the boundaries of the birdcage coil. Black circles and arrows depict peak spatial SAR regions.

REVIEWER #3

7) Do fat suppression techniques interfere with this approach or increase the issue with locally insufficient fat suppression problems in breast imaging?

OUR REPLY

Fat suppression techniques that depend on B_0 homogeneity preserve their quality because B_0 remains the same with the proposed method. In this work, we have demonstrated the performance of the DIXON-based fat suppression method with the resonator as one of the robust fat suppression techniques. A possible limitation might occur only for fat sat and SPIR methods because of their dependence on B_1^+ homogeneity. However, it could be corrected by the RF

pulse shape adjustment (please, see the answer for the comment #3). The techniques that demand high amplitude B_1^+ field (such as SPAIR and STIR) will only benefit from the presence of the resonator.

Added to the main text:

“The ceramic resonator proposed here can be used with all existing MRI sequences, including fat suppression techniques (e.g., DIXON-based⁹). The only necessity could be an adjustment of the excitation RF pulse shapes for some sequences in order to correct the linear excitation RF field inhomogeneity across the breast. The latter has already been successfully realized in the high field clinical 7T scanners⁴¹.

REVIEWER #3

8) Novel sequences (e.g. TWIST VIBE) can acquire images of the full breast in a couple of seconds only, so would it be possible to use your approach to increase resolution instead of reducing imaging time as well?

OUR REPLY

A primary benefit of the proposed here resonator is the substantial reduction of RF power. TWIST-VIBE is a T1w technique that does not demand high amplitude RF pulses (flip angle of only 15 degrees is sufficient). Thus, the proposed resonator will not have prominent benefits for TWIST VIBE. Since the signal-to-ratio of the resonator is comparable to the dedicated receive array, we expect a similar performance of the resonator for TWIST VIBE as with the conventional coils.

REVIEWER #3

9) A “breast coil-shaped” device will be needed for the placement of women in prone position anyhow as well with your device, so what is the "pure" remaining clinical advantage [besides the important factor of costs for the breast coil] – to be asking this as a "provocative" question that might raise at some point when you aim at a clinical introduction of your technique?

OUR REPLY

Indeed, the proposed here experimental setup has several limitations and requires engineering optimization of the design; several clinical advantages are apparent and were discussed in details in the main text:

1. A considerably increased RF safety that can facilitate *in vivo* application of the promising advanced MR sequences with optimal settings to reach a clinically valuable contrast currently shown only *ex vivo* and/or on the phantoms. That is currently not feasible because it is impossible to run these sequences with optimal settings while remaining within the RF safety limits of the birdcage body coil.
2. Reduction of motion-related artifacts because of the focused MR signal excitation.
3. Patient positioning and study planning are simplified and could be automated because the MR signal comes mainly from the region of interest.

Revised in the main text:

“The proposed targeted approach allows for the safer *in vivo* application of these high amplitude RF pulses because of the substantially reduced peak SAR values (see Table S1 in Supplementary Information), which creates optimal conditions for the clinical applications of the advanced MR sequences”.

“Next to it, the absence or a minimal excitation of the other body areas has additional advantages for targeted clinical MRI compared to the conventional MR examinations. Moving organs, such

as a heart, for example, due to the MRI acquisition principles, often create artifacts in the regions of interests, and imaging strategies should be carefully selected to avoid such artifacts. In the targeted MRI, these artifacts are minimized because hardly any signal is excited beyond regions of interest. Hence, the planning of an MR examination is easier and more robust to possible artifacts from the surrounding tissues (see Supplementary Fig. 3)".

"Moreover, the proposed here targeted approach benefits the clinical application advanced techniques, e.g., diffusion-weighted imaging (DWI)⁸ and chemical exchange saturation transfer (CEST)⁹. Both these methods rely on the high amplitude RF pulses and are often performed in vivo under suboptimal conditions because of the RF safety concerns^{10,11}."

REVIEWERS' COMMENTS third round:

Reviewer #3 (Remarks to the Author):

I thank the authors for addressing the questions raised and including further clinically relevant aspects and limitations into the manuscript.

REVIEWER #3

I thank the authors for addressing the questions raised and including further clinically relevant aspects and limitations into the manuscript.

OUR REPLY

We are grateful to the Reviewer for a positive assessment of our work.